# Using Endoscopy in the Diagnosis of Pancreato-Biliary Cancers

**DOI:** 10.3390/cancers15133385

**Published:** 2023-06-28

**Authors:** Julia Chaves, Michael Fernandez Y Viesca, Marianna Arvanitakis

**Affiliations:** Department of Gastroenterology, Hepatopancreatology, and Digestive Oncology, CUB Hôpital Erasme, Université Libre de Bruxelles, 1070 Brussels, Belgium; j.chavesrodriguez@hubruxelles.be (J.C.); michael.fernandez.y.viesca@erasme.ulb.ac.be (M.F.Y.V.)

**Keywords:** endoscopic ultrasound, tissue sampling, intraductal biopsies, pancreatic cystic lesions, cyst fluid analysis

## Abstract

**Simple Summary:**

Endoscopic modalities have a central role in the diagnosis of pancreato-biliary cancers. Endoscopic ultrasound (EUS) is crucial in the diagnosis of both solid and cystic pancreatic lesions through tissue acquisition and fluid sampling. Intraductal brushings and biopsies performed during endoscopic retrograde cholangiopancreatography (ERCP) can provide diagnosis for biliary strictures, and additionally, cholangioscopy can allow direct visualization and image-directed biopsies. Moreover, advances in molecular markers can increase diagnostic accuracy and assist in risk stratification for premalignant lesions, such as pancreatic cystic lesions. The present review focuses on recent developments in the field of endoscopic modalities for the exploration of pancreato-biliary malignant and premalignant lesions.

**Abstract:**

Pancreatic cancer and cholangiocarcinoma are life threatening oncological conditions with poor prognosis and outcome. Pancreatic cystic lesions are considered precursors of pancreatic cancer as some of them have the potential to progress to malignancy. Therefore, accurate identification and classification of these lesions is important to prevent the development of invasive cancer. In the biliary tract, the accurate characterization of biliary strictures is essential for providing appropriate management and avoiding unnecessary surgery. Techniques have been developed to improve the diagnosis, risk stratification, and management of pancreato-biliary lesions. Endoscopic ultrasound (EUS) and associated techniques, such as elastography, contrasted-enhanced EUS, and EUS-guided needle confocal laser endomicroscopy, may improve diagnostic accuracy. In addition, intraductal techniques applied during endoscopic retrograde cholangiopancreatography (ERCP), such as new generation cholangioscopy and in vivo cellular evaluation through probe-based confocal laser endomicroscopy, can increase the diagnostic yield in characterizing indeterminate biliary strictures. Both EUS-guided and intraductal approaches can provide the possibility for tissue sampling with new tools, such as needles, biopsies forceps, and brushes. At the molecular level, novel biomarkers have been explored that provide new insights into diagnosis, risk stratification, and management of these lesions.

## 1. Introduction

Pancreatic adenocarcinoma (PADC) and cholangiocarcinoma (CC) are life threatening oncological conditions with poor prognosis and outcome. These cancers are frequently diagnosed at a later, inoperable, stage and have low 5-year survival rates of 11.5% and 20.8%, respectively [1,2,3]. PADC is the tenth most common cancer but has the third highest mortality in the United States [3]. Survival depends on the stage of cancer, illustrated by 5-year survival rates for pancreatic cancer which span from 43.9% in cases of resectable disease, to 14.7% in cases of locally advanced disease, and 3.1% in cases of metastatic disease [3]. Unfortunately, only 20% of patients are eligible for surgical resection at the time of diagnosis, which underlines the importance of early diagnosis [4].

Obstacles to timely diagnosis include a potentially indolent clinical presentation, inaccurate serum biomarkers, and low specificity and sensitivity of cross-sectional imaging techniques to detect these lesions at an early stage [4,5]. In addition, CC not only has a poor prognosis but can also be very difficult to palliate with optimal biliary drainage, therefore impeding proper oncological management [6].

Pancreatic cystic lesions (PCLs) are mostly benign entities, but some types have a potential for malignant transformation, making characterization and stratification of these lesions crucial to offer appropriate management and surveillance [7,8]. As PCLs are being increasingly detected via cross-sectional imaging techniques performed even in patients without symptoms, evidence-based recommendations are more important than ever for the clinician [9].

Both endoscopic ultrasound (EUS)- and endoscopic retrograde cholangiopancreatography (ERCP)-based techniques can offer substantial information that can be used for determining the diagnosis of both solid lesions and PCLs, as well as undetermined biliary and/or pancreatic strictures. Additionally, a multidisciplinary approach is required to offer appropriate management and avoid misdiagnosis and unnecessary surgical resections that are potentially related to morbidity and mortality [10,11,12].

The scope of this review is to focus on recent advances in the endoscopic diagnosis of malignant and premalignant pancreato-biliary lesions. This includes potential application of EUS-related modalities, as well as ERCP with intraductal visualization and assessment. Advances in tissue acquisition, both EUS-guided and that obtained during ERCP, are also explored. Finally, developments in the molecular field with new biomarkers and next generation sequencing (NGS) are also discussed.

## 2. EUS Techniques

EUS provides pancreato-biliary imaging that is complementary to cross-sectional imaging for both solid lesions and PCLs. It has been proven that computed tomography (CT) and magnetic resonance imaging (MRI) have less sensitivity in detecting smaller pancreatic solid lesions measuring less than 2 cm compared to EUS [13,14,15]. Novel techniques have been developed in the EUS field to improve diagnostic accuracy, such as elastography and contrast-enhanced EUS (CE-EUS). Furthermore, EUS offers the possibility of acquiring tissue or fluid, which is the cornerstone for decision-making regarding management. New needle designs and tissue-acquisition modalities have been developed that have improved tissue specimen quality and diagnostic yield.

### 2.1. Contrast-Enhanced EUS

CE-EUS is a complementary technique to the traditional B-mode EUS imaging that involves a contrast agent which creates microbubbles in the target tissue area once injected intravenously in order to assess local micro-vascularization [16]. The main parameters evaluated are type of enhancement (hyper-, hypo- or non-enhanced), contrast distribution (heterogeneous or homogenous), and speed of wash-out. CE-EUS can allow for better evaluation of a solid lesion or a cystic lesion with a suspected solid component [17].

Regarding solid lesions, CE-EUS can play an important role in differentiating PDAC from other types of lesions, such as neuroendocrine tumors (pNETs) or inflammatory lesions, as seen in patients with chronic pancreatitis and autoimmune pancreatitis [17]. PDAC appears as a hypo-enhanced, homogenous, or non-homogenous lesion, with a fast wash-out, while pNETs appear hyper-enhanced with a slow wash-out, and inflammatory masses present as hyper- or iso-enhanced [17,18,19]. In a recent meta-analysis, the pooled sensitivity, specificity, and diagnostic odds ratio of CE-EUS for the differential diagnosis of PDAC were 0.91 (95% confidence interval (CI), 0.89–0.93), 0.86 (95% CI, 0.83–0.89), and 69.50 (95% CI, 48.89–98.80), respectively [18]. Although CE-EUS does not seem to have a better diagnostic yield than tissue acquisition, it may help if cytology is inconclusive [20]. Furthermore, CE-EUS can help guide tissue acquisition by targeting the most suspicious component of the lesion and avoiding necrotic areas, biopsies of which may yield inconclusive results [16,17,20].

Regarding PCLs, the cystic wall, septae, and mural nodules are assessed for vascularization with CE-EUS. Cystic pNETs, mucinous cystic neoplasms (MCNs), and intraductal papillary mucinous neoplasms (IPMNs) present with a hyper-enhanced wall, whereas pseudocysts have an avascular wall [16,17]. Furthermore, the mural nodules encountered in MCNs or IPMNs at risk of becoming malignant appear as hyper/isoechoic without a hyperechoic rim, whereas mucus or debris are not enhanced [21,22]. Based on recent recommendations, the presence of an enhancing mural nodule over 5 mm is an indication for surgical resection if the patient is deemed fit for surgery. Therefore, CE-EUS can offer crucial input in this setting [9] (Figure 1).

### 2.2. EUS Elastography

Elastography evaluates tissue stiffness by measuring its elasticity [23]. The compression of a target tissue via the EUS probe produces a displacement of the tissue called “strain”, which correlates with the hardness of the structure and may differentiate between benign lesions (soft tissue) and malignant lesions (hard tissue) [16,23]. Additionally, it can be used to guide the biopsy to the optimal area of the lesion to increase diagnostic accuracy [23].

Qualitative and quantitative methods of measurement have been described [24]. Qualitative differentiation is based on a color distinction in which green, blue and red represent normal, hard and soft pancreatic tissue stiffness, respectively. Nevertheless, this measurement is highly operator-dependent and subjective. A quantitative measure, called the strain ratio, is an objective method of stiffness comparison between the target area and a reference area in a grayscale image [24]. Finally, the strain histogram is a computer-enhanced method for dynamic analysis, where color images are transformed into a grayscale of 256 tones. These two aforementioned quantitative measurements allow a more objective assessment. Interestingly, a meta-analysis did not show any difference in diagnostic accuracy between qualitative and quantitative evaluations, with a pooled sensitivity/specificity of 98%/63% and 95%/61%, respectively [24].

EUS techniques can be combined, and it has been reported that EUS-elastography and contrast-enhanced EUS together can improve the accuracy of the diagnosis [19].

### 2.3. EUS-Guided Tissue Acquisition

Despite all the aforementioned advances, the final diagnosis is still based on histopathological sampling. EUS fine-needle aspiration (EUS-FNA) was initially developed to provide tissue for cytological analysis [25,26,27]. On the other hand, fine-needle biopsy (FNB) provides a larger segment of tissue allowing assessment of the architecture and subsequent histological analysis. This is due to an adapted needle tip design that allows more tissue to be sampled and preserves the architectural structure [27] (Table 1).

Rapid onsite cytopathological evaluation (ROSE) consists of the preparation of cytology slides, staining, and assessment of sample adequacy by a pathologist onsite and directly in the procedure room [25]. Macroscopic on-site evaluation (MOSE) consists of the direct macroscopic evaluation of the core tissue obtained from EUS-FNB by the operator [26].

Overall, the diagnostic yield does not differ between FNA and FNB needles, but it seems that FNB needles provide higher sample adequacy [28,29,30,31]. A recent meta-analysis suggested the non-superiority of 22G FNB needles over 22G FNA, with the only advantage being a similar diagnostic yield as FNA but with fewer passes [28]. Regarding FNB needle tip design, a recent review and network meta-analysis showed that Franseen and Fork-tip needles (new generation FNB needles) significantly outperformed the older reverse-bevel FNB needles [30]. Moreover, a multicenter randomized controlled trial confirmed the noninferiority of EUS-FNB without ROSE compared to FNB with ROSE in solid pancreatic lesions when new-generation FNB needles are used, thus highlighting the benefit of the use of FNB needles when the pathologist is not available [31]. Finally, MOSE has an overall diagnostic yield of 90%, sensitivity of 86.5%, and a specificity of 100% for solid pancreatic lesions and may represent a valid alternative when ROSE is not feasible [26].

Different sampling techniques applied during EUS-guided tissue acquisition have been described [32,33]. The classic technique to obtain tissue sampling is the fanning technique (during a single-needle pass, the endoscopist targets different areas to biopsy). A randomized controlled trial compared EUS-FNB using the fanning technique to CE-EUS-guided FNB and revealed similar rates of diagnostic accuracy for solid pancreatic lesions [32]. A recent network meta-analysis showed that the application of suction (specifically wet suction involving saline infusion through the needle) seemed to provide high rates of adequate samples, although with high blood contamination, compared to “no suction” [33]. Adverse events related to EUS-guided tissue acquisition are rare, but may include acute pancreatitis, infection, perforation, and bleeding, with rates estimated to be 0.5–3% of cases [34].

### 2.4. Fluid Analysis for PCLs

EUS-FNA for fluid aspiration and analysis plays an essential role in determining the type of PCL, and, in particular, whether it has a mucinous component, and therefore malignant potential, in cases of non-contributive cross-sectional imaging [35]. High levels of carcinoembryonic antigen (CEA) and mucin staining are consistent with a mucinous PCL, such as MCN or IPMN [35]. Intracystic glucose measurement, which is easily available and inexpensive, has been studied as an additional diagnostic tool. A recent multicenter study in 93 patients showed that a glucose concentration of ≤25 mg/dL had a sensitivity and specificity of 88.1% and 91.2%, respectively, for differentiating mucinous PCLs, whereas a CEA concentration of ≥192 ng/mL had a sensitivity of 62.7% and a specificity of 88.2% [36]. Furthermore, cyst wall sampling using EUS-FNA may also increase the diagnostic yield [35]. Cuboidal epithelial cells, clear cytoplasm, and excess glycogen can diagnose serous PCLs. The presence of mucin, ovarian-like stroma with a degree of cell atypia, is mainly found with mucinous PCLs, such as MCN (Figure 1, Table 2).

Finally, there is an immediate on-site method to improve the diagnostic accuracy of PCL fluid analysis called the string test. A drop of cystic fluid is placed between the examiner’s fingers and then it is stretched, and the maximal length of mucus is measured. It is considered positive if ≥1 cm string is formed and lasts for ≥1 s [37]. The string test has been shown to have a high positive predictive value for correctly diagnosing mucinous PCLs [37].

The analysis of mutations from fluid-containing DNA is increasingly applied in clinical practice. KRAS and GNAS mutations have a good accuracy for the diagnosis of IPMNs and MCNs, based on a recent meta-analysis [38]. Finally, a recent multi-center prospective study showed that NGS of PCL fluid has a high sensitivity and specificity for differentiating between cystic lesions and advanced neoplasia or pNETS [39]. Combining different markers, such as MAPK/GNAS and P53/SMAD4/CTNNB1/mTOR, increased the sensitivity to 89% and specificity to 98% for the diagnosis of advanced neoplasia [39].

### 2.5. EUS-Guided Needle Confocal Laser Endomicroscopy

Needle-based confocal laser endomicroscopy (nCLE) is a novel technique that uses EUS to guide a thin CLE probe through a 19-gauge EUS needle, allowing evaluation of the inner walls of PCLs [8], real-time imaging of intracystic epithelium within a single plane, and in vivo pathological analysis [22].

In a recent prospective observational study, it was shown that the addition of EUS-nCLE to EUS-FNA improved the specificity, sensitivity, and diagnostic accuracy for PCLs [40] as well as an increased diagnostic yield compared to EUS-FNA alone [40].

Typical nCLE features include papillary projections in IPMNs and the superficial vascular network in serous cystic neoplasm (SCN) [40]. Moreover, a recent study identified two criteria related to dysplasia and malignant degeneration: papillary epithelial thickness and darkness [41]. These worrisome features can help in risk stratification for IPMNs [41].

### 2.6. Through-the-Needle Microforceps Biopsy (TTNB)

Although the main diagnostic tool for characterization of PCLs is cyst fluid analysis, cytology of the liquid and the wall can be also be obtained with EUS-FNA but, unfortunately, this is not sufficient for molecular testing and diagnostic yields remain low [38]. Through-the-needle microforceps biopsy (TTNB) allows biopsies to be obtained from the cyst wall with the aid of a miniforceps that is passed through a 19-Gauge EUS needle under EUS guidance [42]. A meta-analysis and systematic review of TTNB including 11 studies [43] demonstrated that TTNB is a superior diagnostic technique compared to FNA for EUS-guided sampling of PCL walls. The most common adverse events (AE) included post-procedural acute pancreatitis (AP) and mild intracystic bleeding. In a prospective study [44], the feasibility of molecular analysis by NGS via TTNB was assessed in 101 patients. The authors demonstrated that TTNB was superior to cyst fluid analysis for differentiating between mucinous and non-mucinous PCLs with a higher sensitivity and specificity, albeit with a 10% AE rate. In addition to the beneficial diagnostic yield of this technique, the rate and severity of AE are not negligible. Indeed, in another prospective open-label controlled study on 101 patients, Kovacevic et al. [45] reported an AE rate of 9.9%, the majority of which was AP. Among these complications, four were considered severe, and one was fatal. More recently, Facciorusso et al. [46] attempted to identify the risk factors for AE in a retrospective study of 506 patients. The AE rate was 11%, including three patients with AP requiring ICU hospitalization and one patient undergoing surgical necrosectomy. Four independent risk factors were highlighted: the type of cysts (IPMN), the number of passages, the complete aspiration of the cyst and the age (>64 years).

Therefore, this technique must only be selected when the benefit of accurate diagnosis outweighs the potential AE, especially since there has been recent development on the identification of molecular markers in the cyst fluid [39,47].

## 3. Intraductal Biliopancreatic Techniques

Biliary strictures are classified as distal when the common bile duct is involved, and proximal when located at the level of the hepatic hilum and intrahepatic ducts [48,49]. Common causes of malignant distal biliary stricture are PDAC followed by CC and ampullary cancer [48,49]. Proximal malignant strictures are mostly related to CC, hepatocarcinoma, gallbladder cancer, and compression due to metastatic lymph nodes [49]. Distal strictures related to PDAC with a mass can be explored with EUS-guided tissue acquisition; however, proximal strictures with no clear mass, as frequently encountered in patients with CC, are more challenging [49].

Biliary strictures are considered as indeterminate when the diagnosis is unclear after cross-sectional imaging and ERCP with biliary sampling. Determining diagnosis is crucial to avoid unnecessary high-risk surgeries as well as a progression to an advanced stage cancer [50,51] (Figure 2).

### 3.1. Intraductal Tissue Acquisition

During ERCP, it is possible to obtain tissue from strictures, under fluoroscopy guidance, via brush cytology and forceps biopsy [50,51]. The yield for brushings varies from 40% to 80%, and can be increased when combined with forceps biopsy [50]. A recent randomized trial confirmed that EUS-FNA had superior accuracy compared to combined brush cytology and forceps biopsy (94% vs. 62%, *p* = 0.003) in cases of extraductal lesions larger than 1.5 cm, but accuracy was similar when considering intraductal lesions less than 1.5 cm [52]. Immunohistochemical staining is a widely available method for investigating specific tumorigenesis-related protein expression patterns in brush and biopsy samples [53,54].

An endoscopic scraper has been developed with a wire-guided system and three scraping loops to obtain tissue and cell samples for histology and cytology. A recently published study including 435 patients with biliary strictures showed that the diagnostic performance of the endoscopic scraper combined with the cell block is better than brush cytology alone or brush with cell block [55]. Nevertheless, sensitivity does not exceed 53%, highlighting the need for complementary investigations.

Malignant biliary strictures lead to chromosomal alterations which can be detected using specific techniques, such as fluorescence in situ hybridization (FISH) and NGS [54]. FISH uses fluorescently labeled complementary DNA probes that allow detection of aneuploidy of chromosomes in biliary brushings or biopsies in order to distinguish between CC and benign bile duct strictures [54,56,57]. Addition of FISH and mutational analysis can increase the level of sensitivity from 32% to 73% for the detection of malignancy, and reach 100% specificity [56]. Nevertheless, overall sensitivities obtained via this technique vary from 31% to 88% according to studies, with a better yield for primary sclerosing cholangitis (PSC)-related strictures and CC [54,56,57].

NGS can detect chromosomal mutations, even in small amounts of tissue or fluid. In a recent study including 252 patients and 346 biliary specimens, the authors identified mutations in a considerable number of biliary brushings and biopsies by performing targeted NGS with a large gene panel [57]. The most prevalent genomic alterations consisted of mutations in KRAS, TP53, CDKN2A, SMAD4, PIK3CA, and GNAS. NGS increased the sensitivity to 83% and maintained a specificity of 99%. The increase in the diagnostic yield was particularly observed for patients with PSC [57].

### 3.2. Cholangioscopy

The development of the single-use, single-operator cholangioscopy device (SOC) that allows direct visualization of the bile tract and targeted biopsies has replaced the previous “mother-baby” peroral cholangioscopy (POCS) system, a device which required two operators and had significant fragility [58].

The first-generation SOC was a fiberoptic device that was replaced by the digital version with improved high-resolution imaging, dedicated aspiration and irrigation channels, and an operating channel that allows the passage of a microforceps to perform biopsies [59,60].

A recent meta-analysis including 13 studies and 876 patients reported an overall sensitivity and specificity of 88% and 95%, respectively [60]. Subgroup analysis showed that SOC image impression provided higher sensitivity but lower specificity than SOC-guided tissue diagnosis with the forceps biopsy.

Direct cholangioscopy can also be applied with ultra-thin endoscopes through direct insertion in the bile ducts. Although this is a challenging procedure, this system may offer the potential of digital chromoendoscopy, like narrow-band imaging, which may increase visualization quality and differentiation of surface structures and architecture [61].

Concerning adverse events, SOC has higher rates of cholangitis related to the need for intraductal perfusion, therefore, the use of prophylactic antibiotics during the procedure is required [58].

### 3.3. Probe-Based Confocal Laser Endomicroscopy

Probe-based confocal laser endomicroscopy (pCLE), also known as optical biopsy, is an endoscopic technique that provides real-time magnification of 1000× microscopic tissue information to diagnose indeterminate biliary strictures [62].

A recent meta-analysis including 18 studies showed that pCLE had a higher sensitivity but lower specificity than tissue sampling during ERCP for the diagnosis of indeterminate biliary strictures [62]. Nevertheless, correct interpretation of the real-time microscopic images can be challenging. Consequently, classification systems have been developed to differentiate between the patterns [63]. A recent prospective study focused on patients with primary sclerosing cholangitis reported high sensitivity for diagnosis of CC, especially at the level of the bifurcation; nevertheless, technical aspects of the probe may limit evaluation of the common bile duct [64]. Major limitations of generalizing the use of pCLE include availability, cost, and lack of expertise.

### 3.4. Pancreatoscopy

Compared to cholangioscopy, peroral pancreatoscopy (POPS) allows direct visualization of the main pancreatic duct (MPD) and tissue acquisition under visual control [65]. Access to the MPD occurs through the major papilla, with or without sphincterotomy, depending on the diameter of the pancreatic orifice and the indication (example, fish mouth encountered in patients with main duct IPMN) [66]. The pancreatoscope is advanced in the duct on a guidewire under regular irrigation and fluoroscopy [65,66]. Therefore, prophylactic antibiotics are recommended in patients undergoing pancreatoscopy because there is a risk of bacterial translocation by irrigating the pancreatic duct with saline solution [65,66]. The two recognized indications are the diagnostic assessment of IPMN (diagnosis, localization, and extension of the disease before surgery) and secondarily for the evaluation of indeterminate pancreatic strictures to discriminate between benign or malignant etiology [65]. Hara et al. first classified pancreatoscopy findings according to the pit-pattern to differentiate benign and malignant aspect with a good accuracy (88% in main duct IPMN and 66% for branch duct IPMN) [67]. A recent meta-analysis (25 studies) showed an excellent diagnostic yield in the diagnostic work-up of IPMN (88–100%). The disease extent of IPMN changed the surgery in 13–62% of the patients. The reported AE event rate was 12%, majority of which was acute pancreatitis (most mild and moderate) [68]. Future studies are needed to better define the role of POP in the diagnostic work-up of IPMN. There are few data regarding the assessment of indeterminate pancreatic strictures when conventional imaging techniques are sometimes insufficient to distinguish between benign and malignant strictures, particularly in patients with chronic pancreatitis [66]. The classification between benign and malignant can be challenging, especially when there is no associated lesion. El Hajj et al. [69] highlighted the role of pancreatoscopy to evaluate pancreatic duct strictures and ductal dilation with different lesions as adenocarcinoma, main and branch duct IPMN and inflammatory strictures. The overall accuracy of visual assessment via POP was 87%, which increased to 94% when pancreatoscopy-guided tissue acquisition was performed [69]. Therefore, current data suggest that in selected patients, pancreatoscopy may play an essential role in characterizing indeterminate pancreatic duct strictures and mapping IPMN before surgery. Nevertheless, pancreatoscopy should be reserved for specific groups of patients due to a narrow range of advantages that this technique allows and also because they can be performed only at expert centers.

## 4. Conclusions and Future Directions

The detection of pancreato-biliary lesions is rising due to an increased use of cross-sectional imaging, even in patients without symptoms. Management of these lesions is crucial due to the potential for malignant transformation. Misdiagnosis can lead to development of advanced neoplasia or unnecessary surgery. Advances have been made in the field of EUS, ERCP, cholangioscopy, as well as in biochemical and molecular detection, to improve diagnosis, risk stratification, and management of these lesions. However, there is a need for prospective, multicenter studies to provide evidence and establish standard guidelines for diagnosis and overall management.

## Figures and Tables

**Figure 1 cancers-15-03385-f001:**
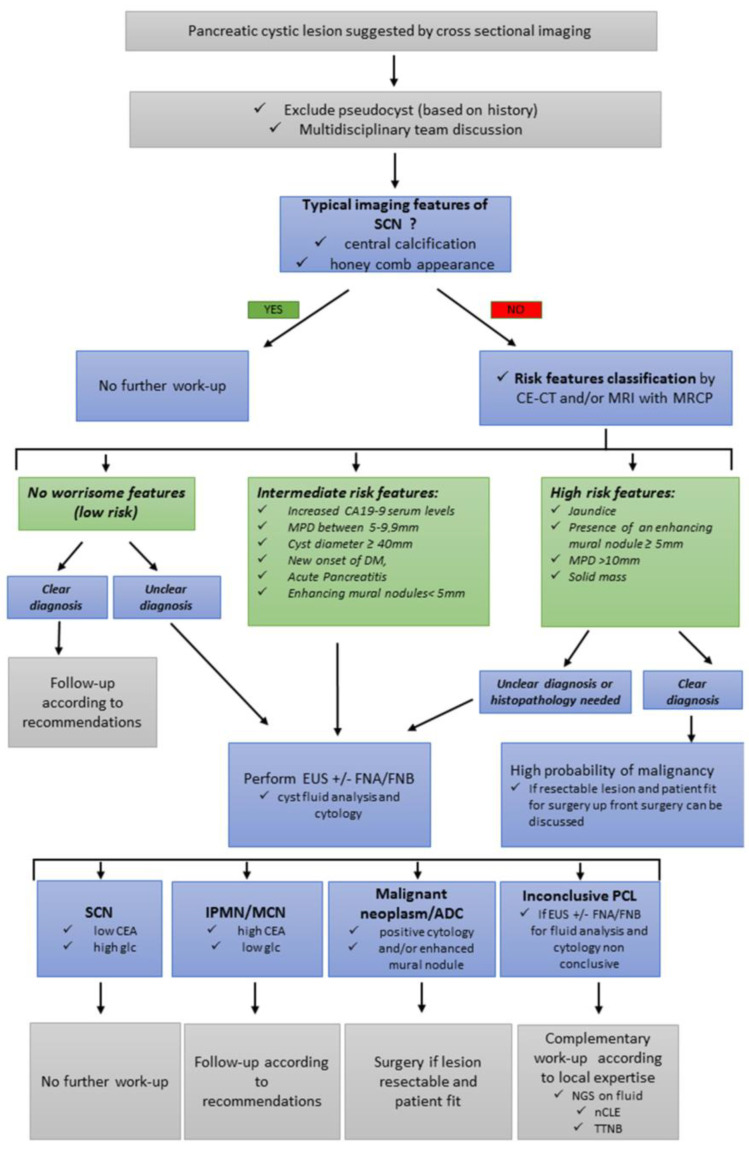
Algorithm for the assessment and management of PCLs. Abbreviations: ADC: adenocarcinoma; CA19-9: carbohydrate antigen 19-9; CE-EUS: contrast-enhanced endoscopic ultrasound; DM: diabetes mellitus; EUS-FNA: EUS fine-needle aspiration; EUS-FNB: EUS fine-needle biopsy; Glc: glucose; IPMN: intraductal papillary mucinous *neoplasm*; MCN: mucinous cystic neoplasm; MPD: main pancreatic duct; PCLs: pancreatic cystic lesions; SCN: serous cystic neoplasm; nCLE: needle-based confocal laser endomicroscopy; TTNB: through-the-needle microforceps biopsy.

**Figure 2 cancers-15-03385-f002:**
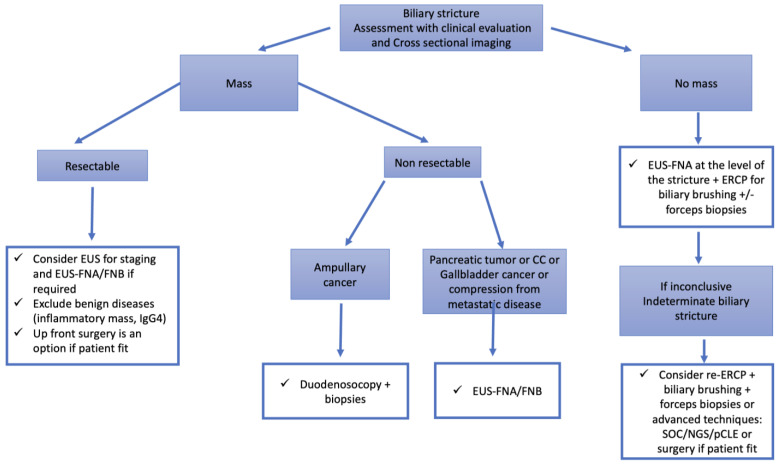
Algorithm for the assessment of biliary strictures. Abbreviations: CC: Cholangiocarcinoma; pCLE: probe-based confocal laser endomicroscopy; ERCP: endoscopic retrograde cholangiopancreatography; EUS: endoscopic ultrasound; FNA: fine-needle aspiration; FNB: fine-needle biopsy; NGS: next generation sequencing; SOC: single-operator cholangioscopy.

**Table 1 cancers-15-03385-t001:** Characteristics of EUS-FNA and EUS-FNB needles.

Needle Type		Characteristics
FNA	Conventional needles (19G, 22G, 25G)	End-cutting needle. Sharply pointed tip to facilitate puncture.
	Menghini-tip needle	End-cutting needle. Tapered bevel edge that facilitates the tissue being withdrawn into the lumen.
FNB	Franseen needle (22G, 25G) *	End-cutting needle. Crown tip with three-plane symmetric cutting edges. No side-slot.
	Reverse-bevel needle (19G, 22G, 25G)	Modified Menghini-type needle with a beveled side-slot near the needle tip. Tissue collected during retrograde movement of the needle.
	Fork-type needle (19G, 22G, 25G) *	End-cutting needle. Fork-shaped distal tip including six cutting edges and an opposing bevel. No side-slot.
	Antegrade core trap (20G)	Modified Menghini-type needle with a beveled side-slot near the needle tip. Tissue collected during antegrade movement of the needle.

Abbreviations: EUS: endoscopic ultrasound; FNA: fine-needle aspiration; FNB: fine-needle biopsy. * new generation FNB needles.

**Table 2 cancers-15-03385-t002:** Different predictors in cyst fluid analysis of PCLs.

	Cyst Fluid Analysis	Mucinous PCLs	Serous/Non Mucinous PCLs	Advanced Neoplasia (Predictors of Degenerate PCLs)
Biomarkers	Intracystic glucose	↘	↗	↘
	CEA	↗	↘	↗
Cytology	Mucin staining	↗	/	+
	Cuboidal epithelial cells	/	+	/
	Clear cytoplasm	/	+	/
	Excess glycogen	/	+	/
NGS	KRAS mutation	+	/	+
	GNAS mutation	+	/	+
	MAPK/GNAS	/	/	+
	P53/SMAD4/CTNNB1/mTOR	/	/	+

Abbreviations: CEA: carcino embryonic antigen; PCLs: pancreatic cystic lesions; NGS: next generation sequencing; Arrow up: increase, Arrow down; decrease, +: presence, /: absence

## Data Availability

Not applicable.

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
