# Peer review of "Using Endoscopy in the Diagnosis of Pancreato-Biliary Cancers"

_cancers, 2023, doi:10.3390/cancers15133385_

Round 1

Reviewer 1 Report

I enjoyed reading the review article on the current possibilities of endoscopy in pancreatobiliary malignancies. The article is well written, comprehensive and give the reader a clear insight into the current reach and limitations of endoscopic methods in this indication.

Author Response

Thank you for your comment. 

Reviewer 2 Report

Very well written and comprehensive review. I have only some minor comments:

1) Figure 1 is quite confusing and not easy to read. The authors should improve it.

2) The authors should mention the potential risk of severe adverse events with TTNB of PCLs. In this regard the authors should comment the two important studies on this topic published recently in Endoscopy (PMID: 35451041 and PMID: 32693411)

Author Response

Thank you for your valuable comments and suggestions. 

We changed Figure 1 according to your comment. We hope it is now more clear and easily readable (page 6).

We also added a paragraph concerning potential severe AE with TTNB according to the two papers you mentioned (pages 9 and 10).

Reviewer 3 Report

a very interesting paper on a very delicate issue

pancreatic and biliary cancers are of different origin even if the methods of investigations are quiet similar 

of course prospective studies are required

Author Response

Thank you for your comment.